# Hypoglossal Nerve Stimulator Placement for Pediatric Trisomy 21 Patients with Refractory Obstructive Sleep Apnea: A Case Series

**DOI:** 10.3390/children7080081

**Published:** 2020-07-24

**Authors:** Joelle B. Karlik, Nikhila Raol, Laura Gilbertson

**Affiliations:** 1Department of Anesthesiology, Emory University School of Medicine, Egleston Children’s Hospital of Children’s Healthcare of Atlanta, 1405 Clifton Drive, Atlanta, GA 30307, USA; laura.gilbertson@emory.edu; 2Department of Otolaryngology, Emory University School of Medicine, Egleston Children’s Hospital of Children’s Healthcare of Atlanta, 1405 Clifton Drive, Atlanta, GA 30307, USA; nikhila.p.raol@emory.edu

**Keywords:** hypoglossal nerve, trisomy 21, obstructive sleep apnea, implantable neurostimulators, airway management, airway obstruction

## Abstract

Background: Hypoglossal nerve stimulators (HNS) are an increasingly popular form of upper airway stimulation for obstructive sleep apnea (OSA) in adults who are unable to tolerate positive pressure treatment. However, HNS use is currently limited in the pediatric population. Case presentation: We present a case series detailing the anesthetic management of three pediatric trisomy 21 patients receiving HNS for refractory obstructive sleep apnea. The patients tolerated the procedure well and experienced no complications. The average obstructive apnea–hypopnea index (AHI) change was 87.4% with the HNS. Conclusions: Proper anxiolysis, safe and controlled induction, multimodal analgesia, and minimization of post-operative respiratory compromise are all necessary to ensure anesthetic and surgical success. After a tailored anesthetic regimen, proper device placement and close follow-up, our patients had a marked improvement in obstructive symptoms.

## 1. Introduction

Patients with trisomy 21 are predisposed to an increased risk of obstructive sleep apnea (OSA) at baseline with as many as 70% meeting criteria [1]. Tonsillectomy, adenoidectomy, and/or lingual tonsillectomies are often performed in patients with adenotonsillar hypertrophy to improve symptoms. However, the majority of patients with trisomy 21 will continue to have ongoing airway obstruction after surgical intervention [1,2,3,4]. Persistent hypotonia, macroglossia, lingual tonsillar hypertrophy, maxillary hypoplasia, obesity, and hypothyroidism all contribute to the continued airway obstruction [1]. Alternative therapies can include non-invasive positive pressure ventilation (NIPPV), oromaxillofacial surgery, and tracheostomy. Recently, hypoglossal nerve stimulators (HNS) have been proposed as a therapy to avoid cardiovascular complications and/or need for tracheostomy [5].

The HNS (Inspire Medical Systems, Inc., Golden Valley, MN, USA) is one of several implantable devices that aims to improve upper airway obstruction [6]. The procedure involves placing a stimulation lead on the anterior branch of the hypoglossal nerve and a sensing electrode in the intercostal muscles (Figure 1). Once turned on, the device stimulates the tongue protrusor muscles after sensing inspiration, causing tongue protrusion and a decrease in airway obstruction during sleep. Three incisions are required: one at the neck inferior to the mandible for the stimulator electrode, one at the anterior chest inferior to the clavicle for the generator pocket, and one at the lateral chest approximately at the fifth rib for the sensor electrode [7]. Both electrodes are then tunneled into the generator pocket for connection. The results from the Stimulation Therapy for Apnea Reduction (STAR) trial, one of the initial multi-center trials in upper airway stimulation in adults, showed that both the apnea–hypopnea index and oxygen desaturation index were decreased by over 67% [8,9]. In adults, the HNS is now Federal Drug Administration-approved for the treatment of OSA in patients with an apnea–hypopnea index (AHI) of > 20 to ≤ 65 events/hours with an age of 22 or older [6]. 

A previous case series of twenty pediatric trisomy 21 syndrome patients receiving HNS has been published examining post-operative efficacy and safety [10]. However, no current publications exist addressing the anesthetic management of this complex patient population receiving an investigational technology. 

We present the anesthetic management of the first three patients receiving an HNS at our institution. This manuscript adheres to the applicable EQUATOR guideline. 

## 2. Case Presentations

Three patients, two female and one male, underwent hypoglossal nerve stimulator placement from 2017–2019 at Egleston Children’s Hospital of Children’s Healthcare of Atlanta as part of trial NCT02344108, “A Pilot Study to Evaluate the HNS in Adolescents with Down Syndrome and OSA.” Currently, HNS use in pediatric populations is still investigational under the Federal Drug Administration. All legal guardians for the subjects gave their informed consent for inclusion prior to participation in this case series.

Baseline demographic characteristics are presented in Table 1. All of the patients had trisomy 21. Comorbidities included asthma (two patients), seasonal allergies (two patients), hypothyroidism (two patients), previous cardiac history (one patient) and seizures (one patient).

All patients were diagnosed with severe refractory OSA despite previous surgery and non-invasive positive pressure ventilation (NIPPV). The previous surgeries included tonsillectomy and adenoidectomy (two patients) as well as lingual tonsillectomy with turbinate reduction and adenoidectomy (one patient). Despite surgical intervention, all patients then required NIPPV therapies including continuous positive airway pressure (CPAP) in two patients and bi-level positive airway pressure (BiPAP) in one patient. All patients failed NIPPV due to poor compliance, primarily caused by inability to tolerate the CPAP or BiPAP device. Each patient underwent a drug induced sleep endoscopy prior to HNS to characterize their residual OSA. Table 1 displays baseline polysomnography (PSG) findings prior to HNS.

Two patients received premedication with oral midazolam for pre-operative anxiety (See Table 2). One patient was not pre-medicated due to a history of respiratory depression with midazolam. All patients underwent an uneventful inhalation induction prior to intravenous line placement. There was no reported difficulty with bag-mask ventilation. All patients were a grade 1 Cormack–Lehane view on direct laryngoscopy. 

The average case length was 244 min (range 227–260 min). Intra-operative courses were overall uneventful. Glycopyrrolate was used for secretions in two patients. One patient received bolus doses of 0.32 mcg/kg epinephrine and 0.08 mg/kg ephedrine for transient bradycardia. Multimodal analgesia was used in all patients (Table 2) and all patients received local anesthetic from the surgeon. Post-op nausea and vomiting (PONV) prophylaxis was used in all patients (Table 2).

All patients were extubated “deep” during stage 3 of anesthesia, transferred to the post-anesthesia care unit (PACU), and placed on blow-by oxygen supplementation. One patient required post-operative oral airway placement, and one patient was placed on his home NIPPV. One patient required post-operative fentanyl and dexmedetomidine for residual pain in PACU. There was no reported post-operative nausea or vomiting. All patients were transferred to the floor for post-operative observation. No anesthetic complications were noted on follow-up. There were no reported complications to the Federal Drug Administration as no serious adverse events occurred at our institution. On follow-up PSG, all patients had notable improvement (see Table 3). The average obstructive AHI change was 87.4% with the HNS.

## 3. Discussion

This case series specifically addresses the anesthetic management of HNS implants in pediatric trisomy 21 patients with refractory OSA. The patients tolerated the procedure well despite their pre-existing conditions, refractory OSA, and relatively prolonged case time. The unique nature of anesthetic management for HNS placement is in the balance between addressing the high-risk patient and operative factors with minimizing respiratory depression and adequately controlling pain. While the majority of our recommendations follow known anesthesia tenants for the care of trisomy 21 patients with OSA, we feel like our recommendations vary in three major ways: the use of premedication, inhalational induction, and deep extubation.

Providing appropriate anxiolysis for parental separation in this patient population is essential. To balance preexisting OSA with the need for preoperative anxiolysis, we found judicious midazolam premedication beneficial in certain patients. Parental presence inductions or other sedative premedications are reasonable alternatives. However, caution should be exercised when administering any sedative medication to this patient population due to theoretical concern of airway obstruction. When sedative premedication is used, either continuous pulse oximetry or direct observation by the anesthesia team is recommended. 

Inhalational inductions are common in pediatric anesthesia in the United States and are to be used with caution in patients with a high risk of airway obstruction. Airway obstruction during stage 2 of an inhalational induction can limit oxygenation and ventilation, especially in a patient with known OSA. However, patients with trisomy 21 may benefit from an inhalational induction as it avoids awake intravenous line placement, anxiety, and fear of medical procedures. Our patient population had minimal obstruction with inhalational induction despite their refractory OSA. We believe that a smooth and controlled inhalational induction is a safe alternative in these patients. Certain equipment should be immediately available including nasopharyngeal and oropharyngeal devices, CPAP capabilities, intubating materials, and intramuscular doses of emergency medications including paralytic.

The intra-operative components attempt to mitigate patient and surgical factors to optimize postoperative patient comfort and safety. We found that the use of primarily short-acting opioids such as fentanyl, multimodal pain control with dexmedetomidine and ketorolac, as well as local anesthetic injected by the surgeon at an insertion site provide satisfactory pain control while avoiding respiratory depression. Other key factors we have recognized include utilizing short-acting inhaled anesthetics such as sevoflurane to ensure timely emergence after deep extubation. We believe that low-dose dexamethasone can decrease the risk of postoperative tongue edema while also synergistically working with ondansetron to prevent post-operative nausea and vomiting, which has the potential to result in subsequent lead dislodgement [11]. 

Other specific anesthetic considerations include placement of monitors and lines opposite of the operative side to avoid complications with use and secure endotracheal tube taping as the bed is turned 180 degrees. Neuromuscular blocking agents are avoided and bite block placed due to intra-operative nerve monitoring. Eye lubrication and careful pressure point padding is required given the case length. Given that sensing lead placement may be complicated by violation of the pleural space, nitrous oxide should be avoided once induction is complete.

Despite the risk of postoperative airway obstruction, operative factors make extubation under a deep plane of anesthesia preferable. The coughing and bucking associated with awake extubation has the potential to cause significant impairment such as a post-operative airway hematoma or HNS lead dislodgement. While no surgical complications occurred in our group, a previous case series on HNS cite a surgical complication risk up to 10% [10]. Our goal with a safe and controlled deep extubation is to minimize such complications and need for repeat surgery. Our limited experience has shown that with a spontaneously ventilating patient, and post-operative NIPPV availability, these patients can be extubated safely under a “deep” plane of anesthesia. To maximize safety while minimizing risks, we recommend close monitoring in the operating room by the anesthesia team after deep extubation until arousal. In a patient with severe obstruction during induction, we suggest discussing deep extubation with the surgical team and consideration of an emergence under light sedation including lidocaine bolus, low-dose remifentanil infusion, and/or dexmedetomidine.

None of the patients experienced anesthetic complications despite their refractory OSA. There were no airway difficulties or significant post-operative respiratory depression that was not able to be managed with supplementary oxygen or home NIPPV. One patient experienced transient bradycardia without hypotension that was not associated with inhalational induction. While bradycardia is common in trisomy 21 patients during inhalational induction, this bradycardia was not thought to be related due its timing in the maintenance phase of anesthesia [12]. The occurrence was prior to incision and may have been due to patient predisposition and low stimulation. 

Patient selection is an important consideration for success of this procedure. Appropriate patients should have high functional capacity with responsible and involved caretakers. As the device requires on/off capabilities, patients should be monitored during use of the device. Close follow-up immediately after implantation can identify early problems and maximize successful use. Our patients had excellent response to the HNS placement with an improvement in the obstructive AHI, meeting or exceeding previously reported percentages [9,13].

## 4. Conclusions

Pediatric trisomy 21 patients with refractory sleep apnea who present for HNS are a growing population [10,13]. Proper anxiolysis, safe and controlled induction, multimodal analgesia, and minimization of post-operative respiratory compromise are all necessary to ensure anesthetic and surgical success. After a tailored anesthetic regimen, proper device placement and close follow-up, our patients had marked improvement in obstructive symptoms. In our experience, anesthetic HNS placement is an extremely rewarding case with a significant lifelong impact in a particularly complicated but gratifying patient population.

## Figures and Tables

**Figure 1 children-07-00081-f001:**
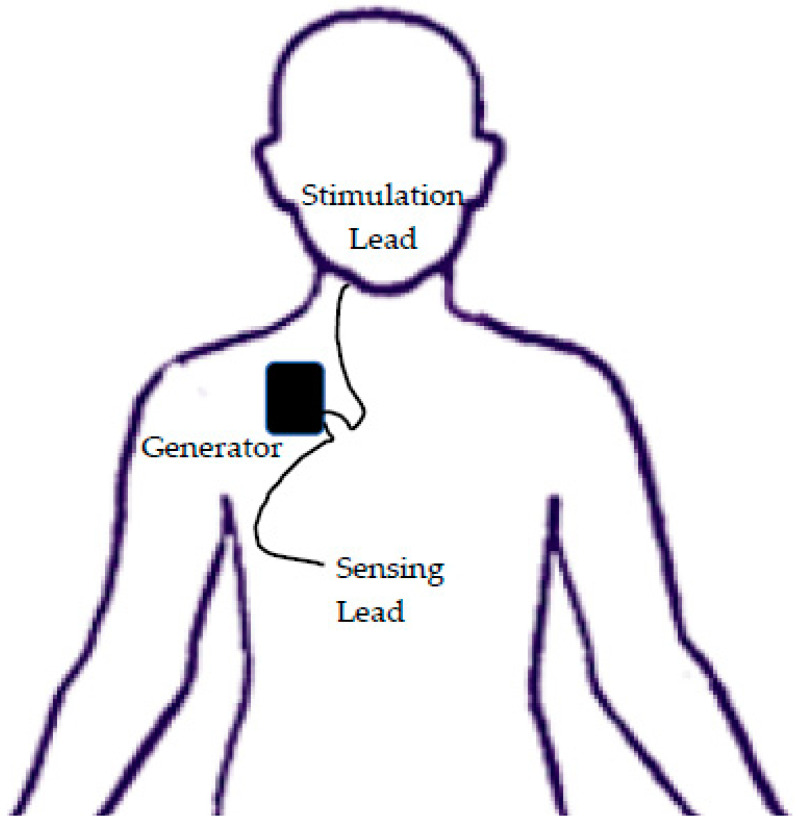
Hypoglossal nerve simulator placement involves a generator, a stimulation lead on the anterior branch of the hypoglossal nerve, and a sensing electrode is placed in the intercostal muscles.

**Table 1 children-07-00081-t001:** Demographics and Baseline Polysomnography.

**Patient**	**Demographics**				
	Age (year)	Weight (kg)	Trisomy 21 Body Mass Index Percentile ^a^	Sex	American Society of Anesthesia Status
**1**	10	36.7	50–75%	F	2
**2**	19	62.1	50–75%	F	3
**3**	13	69	90–95%	M	3
**Patient**	**Baseline PSG**				
	Total Apnea–Hypopnea Index (AHI)	Obstructive AHI	Central AHI	Total Pulse Oximetry Nadir	End Tidal Carbon Dioxide Maximum
**1**	35	35	0.14	78	N/A
**2**	36	34.3	1.19	83	53
**3**	19.7 ^b^	19.7	0	85	55

^a^ BMI Percentiles obtained at: https://pediatrics.aappublications.org/content/pediatrics/early/2015/10/21/peds.2015-1652.full.pdf. ^b^ Total apnea–hypopnea index (AHI) recorded during Continuous Positive Pressure titration.

**Table 2 children-07-00081-t002:** Anesthetic plan and medications.

	**Midazolam Premedication**	**Inhalation Induction**	**Sevoflurane for Induction**	**Nitrous Oxide for Induction**	**Paralytic Used for Intubation**	**Sevoflurane for Maintenance**	**Paralytic Used for Maintenance**
**Patient 1**	0.41	yes	yes	no	no	yes	no
**Patient 2**	0	yes	yes	yes	no	yes	no
**Patient 3**	0.22	yes	yes	no	no	yes	no
	**Fentanyl mcg/kg**	**Hydromorphone mg/kg**	**Ketorolac mg/kg**	**Dexmedetomidine mcg/kg**	**Glycopyrrolate mg/kg**	**Decadron mg/kg**	**Zofran mg/kg**
**Patient 1**	1.82	0.004	0	0.65	0.0027	0	0.11
**Patient 2**	3.22	0	0	0.84	0	0.13	0.06
**Patient 3**	1.45	0	0.434	0.29	0.0029	0.09	0.08

**Table 3 children-07-00081-t003:** Change in obstructive apnea–hypopnea index (AHI) after hypoglossal nerve stimulators (HNS) placement.

	Initial Obstructive AHI	Post-HNS Obstructive AHI	Percentage Change
**Patient 1**	35.0	2.8	92.0
**Patient 2**	34.3	5.9	82.8
**Patient 3**	19.7 ^a^	2.5 ^b^	87.3

^a^ Obstructive AHI recorded during Continuous Positive Airway Pressure titration. ^b^ Obstructive AHI recorded during optimal positioning.

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
