# Peer review of "Hypoglossal Nerve Stimulator Placement for Pediatric Trisomy 21 Patients with Refractory Obstructive Sleep Apnea: A Case Series"

_children, 2020, doi:10.3390/children7080081_

Round 1

Reviewer 1 Report

Dear Authors: Thank you for a well designed study investigating the role of upper airway stimulation in patients with OSA and Trisomy 21. 

I have included several comments in the attached PDF.

A few major points:

  • Please clarify decision process around induction in patients with severe OSA (likely difficult airways) without IV access
  • Please clarify decision process involving deep extubation - again in patients with known severe OSA who are likely to obstruct and may be challenging to reintubate. Several considerations can be made for a smooth emergence (use of lidocaine, remifentanil, etc).
  •  Please cite the STAR trial and follow up discussing the benefit of upper airway stimulation
  •  Please highlight upon the decision making process of placing the HNS in patients with developmental delay. As these device require the user to turn them on and off appropriately. Are the patients able to do this or are the providers doing so? And is there risk of these devices being harmful in this patient population?

Reviewer 2 Report

When presenting the information about using multimodal analgesia (during the case information portion), it may be useful to make a table. The paragraph, though brief, has a lot of information in it (i.e. ALL patients received some medications, while only 1 patient each received other medications). This can be confusing to read in paragraph form. In addition, as you continue on to the Discussion section, the reader learns that [some] patients received IV acetaminophen and local anesthetic [from the surgeon] as well. A table would likely help efficiently relay this information.

May want to flesh out that the conclusion a bit to really wrap everything up.

Overall, the paper is well-written.

Reviewer 3 Report

1.) For those not familiar with the actual implantation procedure for a HNS it would be helpful to point out briefly what the surgical procedure entails-location and extent of incisions, how the electrodes are tunneled or accessed to their sites. This helps to provide a better understanding of what one might be thinking about at baseline in regards to what an anesthetic regimen or post-op pain regimen would be expected. Then your details about your case series help to show the difference (or similarity) in this patient population. This could either be addressed during the Case Presentations sections or in the Discussion when addressing operative factors that should be considered 

2.) The case data regarding meds given might be better viewed or easy to read in a table format. Being able to see weight based dosing for each patient vs an average of the three. (Lines 83-96)

3.) Was muscle relaxant used to facilitate intubation? Should it be avoided in HNS cases (along the same lines as VNS). A mention of this would be helpful for anesthetic planning.

4.) Was dexamethasone given at PONV dosing or higher dosing because of airway edema concerns? How big of a concern is post-op airway edema due to surgical technique

5.) Was bradycardia requiring treatment during induction or maintenance? Is that when we normally anticipate this? Normal to require treatment with Epi?

Reviewer 4 Report

This case series is a well-written and succinct report of the authors' experience providing safe and appropriate anesthesia for Tri21 patients undergoing HNS placement, however there is not much about this anesthesic management that is particularly unique to this procedure, or particularly special about what the authors did in the management of this case.

The patient population described in this series is one that most pediatric anesthesiologists are very familiar with, and care for often. The issues and considerations mentioned by the authors ("balance between addressing the high-risk patient and operative factors with minimizing respiratory depression and adequately controlling pain") are well known to pediatric anesthesiologists. The principles of management the authors describe (multimodal analgesia, short acting opiates, steroids to combat edeama, short acting inhaled anesthetic agents, judicious deep exutbation, etc) are well established and performed routinely for these particular patients when undergoing many different types of procedures, not something new that has been developed specifically for the placement of the HNS. 

Additionally, there are several points emphasized by the authors in the discussion that are not as well supported by their accounting as they should be. For example, the authors discuss the importance of "multimodal pain control with IV acetominophen" (with which i think most pediatric anesthesiologists would certainly agree) but none of the patients presented in this particular series actually received it. They also specifically point out the "essential" importance of using dexamethasone to combat postoperative airway edema, but this was not actually used in all of their patients, so it is a little bit difficult to claim it is therefore "essential."

One thing that would strengthen the authors submission and would make the paper more publishable would be a greater evaluation of the use of "deep extubation" in this patient population for this procedure. As the authors correctly point out, this is a controversial practice in patients with Tri21 known severe OSA (deep extubation is traditionally contraindicated here). If the authors were to more carefully evaluate this practice in particular, and perhaps do a comparison of hematoma formation secondary to bucking, lead dislodgement with coughing,  PACU issues, post-op problems, etc between  patients who were "extubated deep" and those who were not, they might be able to extend their claim that this is an important aspect of the anesthetic care of these patients and argue that it should be used for this particular procedure and this paper would be more impactful. 

Round 2

Reviewer 1 Report

Thank you to the authors for thoroughly addressing the comments and queries. The changes provide further insight into the strategies utilized and considerations around patients undergoing implantation of upper airway stimulators with Trisomy 21. 

The first publication from the landmark STAR Trial is

  • Strollo PJ, Soose RJ, Maurer JT, et al. Upper-airway stimulation for obstructive sleep apnea. N Engl J Med, 2014; 370(2): 139-49.

Reference 8 is one of the follow up investigations to this original study. Please consider citing this original study as well. 

Thank you.

Reviewer 4 Report

Thank you for your prompt response to reviewer suggestions. I think that by increasing your emphasis on what is novel/controversial about your approach to these patients and the associated evidence for your technique is much improve in this version. 

Author Response

There were no requested changes.